

# High-resolution numerical modelling of seasonal volume, freshwater, and heat transport along the Indian coast

Kunal Madkaiker[1], Ambarukhana D. Rao[1], Sudheer Joseph[2]

[1]Indian Institute of Technology Delhi, New Delhi, India

[2]Indian National Centre for Ocean Information Services (INCOIS), Ministry of Earth Sciences, Hyderabad, India

*Correspondence to*: Kunal Madkaiker ( Kunal.Ajit.Madkaiker@cas.iitd.ac.in)

**Abstract.** Seasonal reversal of winds and equatorial remote forcing influences the circulation of the Arabian Sea (AS) and Bay of Bengal (BoB) basins in the North Indian Ocean. In this study, we numerically modelled the physical characteristics of

AS and BoB, using the MITgcm model at a high spatial resolution of 1/20° forced with climatological initial and boundary conditions. The simulated temperature, salinity, and flow fields were validated with satellite and in situ datasets. We then studied the exchange of coastal waters by evaluating transports computed from the model simulations. The alongshore volume transport on the eastern coast is stronger with high seasonal variability due to the poleward-flowing western boundary current and equatorward-flowing East Indian Coastal Current. West coast transport is influenced by large intraseasonal oscillations.

The alongshore freshwater transport is two orders less than the alongshore volume transport. Out of the net volume transport, freshwater accounts for a maximum of 6.03 % during the southwest monsoon season followed by 4.85 % in the post-monsoon season. We observe an inverse relationship between alongshore freshwater and volume transport on the western coast and a direct relationship on the eastern coast. Seasonal variations between the cross-shore volume transport and its alongshore component also present such a contradiction along the western coast while displaying in-phase behaviour on the eastern coast.

We also observed that meridional heat transport over AS is stronger than BoB. Both basins act as a heat source during the summer monsoon and heat sink during the winter. This high-resolution model set-up simulates all the important physical climatological patterns making it a useful tool in various physical as well as biogeochemical studies in this region.

## 1 Introduction

The Indian coastline is surrounded by the north Indian Ocean (NIO) with the Arabian Sea (AS) on the west of the mainland and the Bay of Bengal (BoB) to the east. The current pattern in the NIO is dictated by the reversal of winds and equatorial remote forcing (Schott and McCreary, 2001; Shankar et al., 2002; Rao et al., 2010). This helps in the exchange of freshwater and thermal ventilation of AS and BoB waters. These waters are unique as both the basins are landlocked from three sides and have proximity to the coasts. Also, the thermohaline properties are different in these basins. AS is a highly saline basin due to

excessive evaporation over precipitation and the advent of high-saline waters from the Persian Gulf and Red Sea (Bower and Furey, 2012; Zhang et al., 2020). BoB is comparatively fresher due to the impact of precipitation and river runoff (Behara and



Vinayachandran, 2016; Amol et al., 2020; Jana et al., 2015, 2018; Srivastava et al., 2022). Modelling studies in these basins help to understand the impact of these unique characteristics over various physical and biogeochemical aspects.

Two important coastal current systems are flowing in these basins. In the AS, the West Indian Coastal Current (WICC) is an eastern boundary current along the west coast with an equatorward (poleward) flow during the summer (winter) season (Shetye et al., 1991a; Shetye and Gouveia, 1998; Shankar et al., 2002). The course of this current also varies interannually. It is strongly seasonal around its central path (Mumbai, ~20°N) coast as compared to the southern (Kollam, ~9°N) coast (Chaudhuri et al., 2020). It becomes wider (narrower) along the southwest (northwest) coast of India (Shetye et al., 1991a). The flow of WICC

is governed by winds as well as remote forcing (Shankar and Shetye, 1997; Shankar et al., 2002; Shetye et al., 2008). In the BoB, a poleward flowing Western Boundary Current (WBC) flows during the pre-summer monsoon season and an equatorward flowing East Indian Coastal Current (EICC) flows during the winter season (Shetye et al., 1996). The WBC is formed due to the wind-stress curl (Gangopadhyay et al., 2013) whereas the EICC is generated as a combined effect of density-driven flow and a coastally trapped Kelvin wave (Rao et al., 2010). These current systems are integral components governing

the transport of volume and freshwater alongshore within the basins. Two monsoon currents namely the southwest monsoon current and the northeast monsoon current are responsible for the transport of water between the AS and BoB. The southwest monsoon current associated with summer monsoon winds, advects saltier water from southeastern AS into southwestern BoB (Murty et al., 1992; Vinayachandran et al., 1999). The northeast monsoon current flows as a combined effect of the coastally trapped Kelvin wave and the westward propagated Rossby wave originated in the eastern equatorial IO (Schott et al., 1994;

Shankar et al., 2002). It transports freshwater from southwestern BoB into southeastern AS. Various studies have attempted to understand the freshwater exchanges in the NIO based on satellite-derived observations (Akhil et al., 2020; Papa et al., 2012; Mahadevan et al., 2016), surface drifters (Hormann et al., 2019) and Argo floats (Parampil et al., 2010; Lin et al., 2019). Using numerical modelling we can additionally account for the subsurface ocean state, to understand the transports at deeper depths. This remains a limitation in surface observational studies.


The MITgcm (Massachusetts Institute of Technology general circulation model) (Marshall et al., 1997) is a numerical model which finds widespread utility in studying diverse ocean-related applications globally (Menemenlis et al., 2005; Forget et al., 2015; Stammer et al., 2003; Mazloff et al., 2010; Gopalakrishnan et al., 2020; Srivastava et al., 2016). Notably, its significance is underscored by the inclusion of various packages, augmenting its physical model to extend to various biogeochemical

processes, thereby advancing our understanding of complex oceanic interplays. The broad goal of this study was to set up the MITgcm model over our domain and estimate surface and subsurface volume, freshwater and heat transports in the basin. For this, we configured the model over AS and BoB (Fig. 1) using climatological initial and boundary forcing data. We initialized the model simulations with precise data from *in situ* and the latest satellite observations to minimize the margin of error. Then, we selected appropriate physical parametrization schemes and determined the optimal values for horizontal, and vertical

viscosity and diffusivity. We also rigorously validated the simulated physical parameters with satellite and gridded Argo





observations. This setup was then used to simulate the coastal circulation and the resulting volume, freshwater and heat transports along the Indian as well as eastern Sri Lanka coastline (Fig. 1) and to understand how it gets modulated by winds, remote forcing, and flow fields. Estimating transports using *in situ* observations in the NIO is challenging due to their sparse coverage in time and space. Satellite derived surface currents too are limited due to their coarser resolution and inability to

capture subsurface data. Thus, employing our validated MITgcm setup for transport computations emerges as a favourable alternative. To ensure the meaningfulness of our simulations in areas where observations are lacking, we conducted a model-to-model comparison of transports using the INC-HYC setup. Our study is vital in taking forward our understanding from existing studies such as Sen et al. (2022), which provides an averaged picture of the volume transports along the WICC and EICC. We also analysed the alongshore flow in our analysis which is a main component of the coastal currents and their

exchanges (Amol et al., 2014). Furthermore, we investigated the AS and BoB meridional heat transport to analyse the heat distribution which helps to understand how heat gets distributed in the NIO during different seasons.

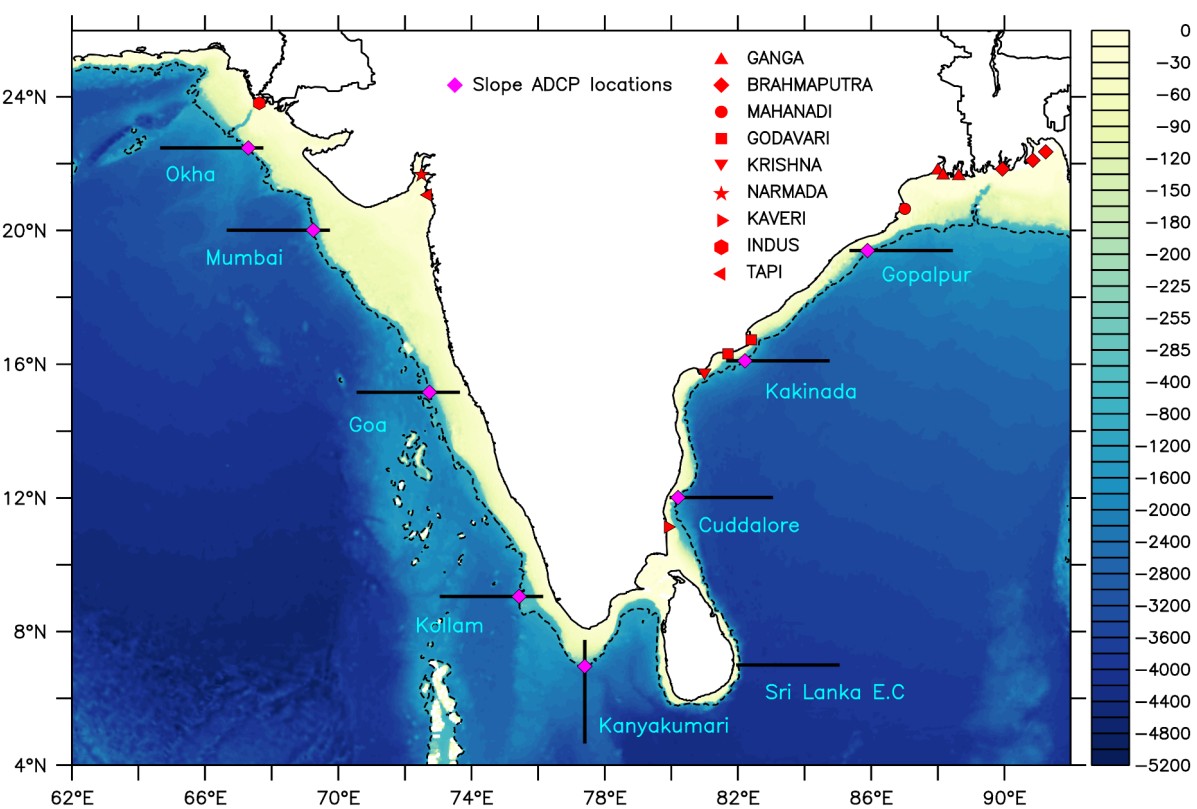

**Figure 1. Study area depicting the bathymetry of the model domain (in colour scale). The dashed black line represents the 1000m**

**bathymetry contour. Red symbols represent the discharge points of major river systems in India. Purple symbols represent the slope ADCP locations, along which the transects have been considered. All transects are 3° in length.**



The resolution of the modelling experiments plays an important role in the accurate forecasting and simulation of mesoscale and sub-mesoscale processes over regional domains (Benshila et al., 2014). We selected a horizontal resolution of 1/20° (~5 km) which allows the simulations to be eddy-resolving. Such a fine resolution is important when considering basins which are close to coasts. This study is further aimed to help researchers assess its implications on nutrients/sediment transport, biological productivity, and the fishery sector. This setup can also be used with appropriate forcings to forecast the physical state of the ocean in this region. We describe the models used for this study, their configuration, the different datasets, and methods used in section 2. In section 3, we discuss the model validation and discussion. Conclusions are presented in Section 4.

## 2 Materials and Methods

### 2.1 Model

### 2.1.1 MITgcm description and configuration

The MITgcm (Marshall et al., 1997) (hereafter model), is configured for a regional domain (62°E - 92°E, 4°N - 26°N) which includes the AS and the BoB. The model solves Navier-Stokes equations by using Boussinesq approximation. Here we have chosen the hydrostatic approximation. The finite volume approach is used over a staggered Arakawa-C grid. The third-order direct space-time advection scheme is used for temperature and salinity. The model setup has a uniform high spatial resolution of 1/20° (~ 5 km), with 49 vertical levels in a z-coordinate system. Such a fine resolution allows the simulations to be eddy-resolving. The vertical resolution is 5m from the surface to 250 m depth, and gradually decreases at a deeper depth. The maximum depth of the model setup is 4500m. Bathymetry is obtained from the General Bathymetric Chart of the Oceans (GEBCO) (Weatherall et al., 2020), having a high spatial resolution of 15-arc-sec. We use K-profile parameterization (KPP) (Large et al., 1994) as the vertical mixing parameterization scheme and MDJWF equation of state (McDougall et al., 2003). There are three lateral open boundaries along the western, eastern, and southern edges of the model domain. No-slip and free-slip boundary conditions are applied at the bottom and lateral boundaries, respectively for velocities. An implicit free surface and a non-rigid lid condition are implemented for the surface pressure. The model is integrated over an optimized timestep of 120 seconds and is designed using climatological forcing data. Initial temperature and salinity are prescribed from the World Ocean Atlas 2018 (WOA18) climatology (Locarnini et al., 2018; Zweng et al., 2019). To reduce the spin-up time and computational expenditure, the model is initialized with a warm start. For this, climatological zonal and meridional currents are prescribed from the Simple Ocean Data Assimilation (SODA) 3.12.2 dataset (Carton et al., 2018). Lateral boundary conditions, that is, temperature, salinity, and currents are also prescribed from SODA 3.12.2 on a pentad climatological time scale. The model is forced with daily climatology of 2m air temperature, 2m specific humidity, 10m zonal and meridional wind, net shortwave, and longwave radiation as the atmospheric forcing, from the fifth-generation reanalysis (ERA5) (Hersbach et al., 2020). Precipitation is obtained from Global Precipitation Measurement level 3 (GPM) (Huffman et al., 2015). Along with this, climatological river discharge from Dai and Trenberth (2002) is fed to the model using a point-source method, to incorporate the effect of river runoff at the locations marked red in Fig. 1. The model uses bulk formulae to calculate net





heat and freshwater flux using the above parameters. The model is relaxed at the surface with monthly climatology of Multi-scale Ultra-high Resolution v4.1 Sea Surface Temperature (MUR SST) (Chin et al., 2017) and Soil Moisture Active Passive Sea Surface Salinity (SMAP SSS) (Tang et al., 2017) satellite data, prescribed every 10th day. The data links of all datasets are provided in the code and data availability section.


### 2.1.2 HYCOM model description

We used the HYbrid Coordinate Ocean Model (HYCOM) (Bleck, 2002) data for performing a model-to-model comparison with the MITgcm simulations. The model is simulated at the Indian National Centre for Ocean Information Services (INCOIS; hereafter referred to as INC-HYC) and is an eddy-resolving model with a high spatial resolution of 1/16° (~ 6.25 km) over the

IO (20°E–120°E, 43.5°S–30°N). It is one-way nested within a global HYCOM setup of 1/4° (~ 25 km) spatial resolution. The model equations are solved on the Arakawa-C grid using the finite volume method. The second-order enstrophy conserving scheme is used to compute momentum advection (Bleck and Boudra, 1986; Bleck and Smith, 1990). Continuity equations and physical tracers such as temperature, and salinity are computed using a second-order flux-corrected transport scheme (Zalesak, 1979; Iskandarani et al., 2005). The model has 29 vertical hybrid layers. This setup incorporates the Tentral Statistical

Interpolation (T-SIS) data assimilation scheme, based on multivariate linear statistical estimation. A combination of GEBCO and ETOPO-1 is used as the model bathymetry. The atmospheric forcings, that is, 2m air temperature, 2m vapour mixture, surface downward and upward long and shortwave radiation, precipitation, and 10 m zonal and meridional wind are obtained from NOAA Global Forecasting System (GFS). Wind stress is computed according to Kara et al. (2005), whereas bulk formulae used to calculate surface fluxes are referred from Kara et al. (2000). KPP (Large et al., 1994) is chosen as the vertical

mixing scheme. Monthly climatological river discharge is used from the Naval Research Lab (NRL). The model is simulated for a five-year hindcast period (2012-2016) and the state variables are written on daily intervals. We have computed daily climatology from this data. The technical report (https://incois.gov.in/documents/TechnicalReports/ESSO-INCOIS-CSG-TR-01-2018.pdf) can be referred to for more details and exhaustive validation. The HYCOM simulated data is reliable and performs reasonably well when compared with other data assimilated products such as NRL-HYCOM and INCOIS-GODAS.


### 2.2 Data

To validate the seasonal model SST and SSS, we used the Group for High-Resolution Sea Surface Temperature (GHRSST) (Donlon et al., 2012) and Soil Moisture and Ocean Salinity (SMOS) SSS v8.0 level 3 (Boutin et al., 2023) datasets respectively. The GHRSST level 4 data by NASA's JPL, is produced using the optimal interpolation method at 0.054° spatial resolution.

The latest version of SMOS uses an improved de-biasing technique and is available at 0.259° x 0.196° spatial resolution. Apart from these satellite datasets, we have also validated the model temperature and salinity with the gridded Coriolis Ocean Dataset for Reanalysis (CORA v5.2) (Szekely et al., 2019) and RAMA buoy (McPhaden et al., 2009) (refer supplementary text S1 and S2). CORA v5.2 data has been produced using multiple *in situ* sampling techniques such as Argo floats, drifters, gliders, moorings, etc. It has a spatial resolution of 0.5°, with data available from the surface to 2000 m depth. GlobCurrent zonal and



meridional surface currents (Rio et al., 2014) were used to validate model computed surface currents. This observational data, derived from satellite mean dynamic topography, is available at the surface as well as 15m, as a sum of geostrophic and Ekman currents. This data has a spatial resolution of 0.25°. We computed monthly climatology for all these datasets and compared them with seasonal patterns of simulations.

## 2.3 Methods

We initialized the model with the December monthly climatology of ocean temperature and salinity from WOA18, and zonal and meridional currents from SODA 3.12.2. Boundary conditions were forced from SODA 3.12.2 for the climatological year, on a pentad time scale. For appropriate selection of boundary conditions, we conducted a few sensitivity experiments using different reanalysis products (SODA 3.12.2, ECCOv4, and INCOIS-GODAS) and observed that SODA 3.12.2 was able to

represent the ocean temperature, salinity, and circulation better in the domain, than others. Similarly, ERA5 wind data was compared with wind from RAMA buoy at various locations over AS and BoB and it was observed that ERA5 shows a strong correlation, lower RMSE and low standard deviation with RAMA. Hence, we selected this dataset for the surface forcings. All forcing datasets were regridded to the model domain resolution. The model was spun up for 5 years. Based on the temporal evolution of kinetic energy, we found that it achieves a steady state after 5 years. The 6[th] year model output was considered

for our analysis. Model outputs were written on a pentad scale. To compute the transports for the western coast of India, we computed the alongshore and cross-shore current components following Shah et al. (2015):

$$\text{acomp} = (v\cos(\theta) - u\sin(\theta)) \tag{1}$$

$$\text{ccomp} = (u\cos(\theta) + v\sin(\theta)) \tag{2}$$

where u and v are the zonal and meridional current components and θ is the angle of rotation (between true north and coast,

anticlockwise (+ve) direction) in radians. Here, acomp and ccomp are the alongshore and cross-shore current components respectively. The angle of rotation for the western and eastern coasts was obtained from Amol et al. (2014) and Mukherjee et al. (2014) respectively. Similarly, on the eastern coast:

$$\text{acomp} = (v\cos(\theta) + u\sin(\theta)) \tag{3}$$

$$\text{ccomp} = (u\cos(\theta) - v\sin(\theta)) \tag{4}$$

The angle of rotation is clockwise (-ve) direction. The alongshore volume transport (AVT) and cross-shore volume transport (CVT) are computed in Sv (1 Sv = 106 m$^3$/s) following Stammer et al. (2003):

$$\text{AVT} = \iint \text{acomp} \, dz \, dx \tag{5}$$

$$\text{CVT} = \iint \text{ccomp} \, dz \, dx \tag{6}$$

Alongshore freshwater transport (AFT, in Sv) is computed referring to Stammer et al. (2003) and Rainville et al. (2022):

$$\text{AFT} = \iint \text{acomp} \, \frac{(s_{ref} - s)}{s_{ref}} \, dz \, dx \tag{7}$$





where S is the ocean salinity (psu) and $S_{ref}$ is the reference salinity of 34.83 psu. It should be noted here that the freshwater fraction is always negative when $S_{ref} < S$ as in the AS, irrespective of the orientation of the alongshore component. Thus, the sign convention of AFT in AS will always be inverse of the volume transport. However, in the BoB, the freshwater fraction

will be mostly positive ($S_{ref} > S$). This makes the direction of AFT to be dependent only on the alongshore flow.

Alongshore heat transport (AHT) is computed in PW (1 PW = $10^{15}$ W) referring to Stammer et al. (2003) and Chirokova and Webster (2006):

$$AHT = \rho C_p \iint acomp \, T \, dz \, dx \tag{8}$$

where T is ocean temperature (°C), $\rho$ is seawater density (kg/m³) and $C_p$ is the specific heat capacity of seawater (3898 J/kg°C).

The net volume transport (NVT), net freshwater transport (NFT), and net heat transport (NHT) were obtained by summing their respective zonal and meridional components (Arora, 2021). The zonal and meridional components were computed following Stammer et al. (2003). To understand the seasonal transport variability, NVT and NFT were computed at each grid

point integrated over the upper 100 m. Based on the transport estimates, we quantified how much freshwater gets advected by WBC and EICC along the eastern coast of India. The freshwater contribution was computed as a percentage of the NVT. The transports computed from the MITgcm model are compared with the same from INC-HYC.

## 3. Results and Discussion


### 3.1. Model validation







**Figure 2. Seasonal climatology of SST (in °C) from (a-d) model, (e-h) GHRSST and (i-l) gridded Argo SST and bias of model with (m-p) GHRSST and (q-t) Argo. R values represent Pearson's correlation coefficients.**

Model validation is a crucial step which ensures that the model accurately represents the complexities of oceanic dynamics. For validation, we compared the model SST (Fig. 2), SSS (Fig. 3) and surface currents (Fig. 4) with observations and computed biases.



As seen from both the observation datasets (Fig. 2e-l), we note some climatological characteristics in the SST distribution
across these two basins. The seasons have been distinguished as pre-summer monsoon (MAM), summer monsoon (JJAS),
post-summer monsoon (ON) and winter (DJF). In MAM, NIO is the warmest and we observe BoB to be warmer than AS. This
is a cumulative effect of heat being stored in the upper few meters of the ocean due to strong stratification and increased surface
net heat flux. The WBC advects warmer waters from the southwest up into the northwestern BoB. During JJAS, the
temperature difference between the South Asian landmass and ocean causes the Intertropical Convergence Zone to shift
northward (Gadgil, 2003). The strong southwesterly winds exceeding the speed of more than 10 m/s cause latent heat to release
in the AS, cooling surface waters. We observe a cold tongue flowing from AS into BoB, also called a 'cold pool', which is
maintained by advection and mixing due to entrainment (Vinayachandran et al., 2020). This coastally upwelled cold water
along the southeastern AS is further advected into BoB by the southwest monsoon current. During the transition season of ON,
wind direction reverses and becomes north-easterly. We observe a warming signal along the western coast of India during this
period. In DJF, we observed SST cool in the northern AS and BoB. Cold and dry winds blowing from the northeast cool the
surface waters by releasing latent heat through evaporation. This along with strong convective mixing causes a reduction in
net heat flux (Kumar and Prasad, 1996). During this period, we observe warmer temperatures along the southeastern AS. This
can be attributed to the westward propagating downwelling Rossby wave, which traps heat beneath the deepened thermocline
and prohibits ventilation (Kushwaha et al., 2022).


The model setup simulates these seasonal SST patterns and compares well with both GHRSST and gridded Argo with an
acceptable bias range. The model has a warmer bias in northeastern AS during JJAS when compared with GHRSST (Fig. 2n)
owing to an increase in net heat flux. In ON, model SST is warmer along the western coast of India, compared to the
observations (Fig. 2o, s). Overall, we note that the model has a slightly warmer bias over parts of northern AS and along the
western coast where the shelf is wide majorly due to shallower water depth in the coastal regions. The heat gets trapped into
the upper few meters, thereby increasing the surface net heat flux. We evaluated the model SST with observations statistically
using a robust parameter, Pearson's correlation coefficient. We observe that simulated SST shows a strong positive correlation
up to even 0.98 with GHRSST and gridded Argo SST throughout the climatological year.





**Figure 3. Seasonal climatology of SSS (in psu) from (a-d) model, (e-h) SMOS and (i-l) gridded Argo SSS and bias of model with (m-p) SMOS and (q-t) Argo. R values represent Pearson's correlation coefficients.**

SSS is another important parameter governing the transports. Similar to SST, climatological SSS has certain characteristic features in NIO. During MAM season, the poleward WBC brings saltier water from the southwestern up into the head BoB region. As the JJAS season commences, the Indian peninsula receives huge amounts of freshwater as a cumulative effect of precipitation as well as the melting of snow in the Himalayan regions. This increases the riverine discharge into the head BoB



making it less saline. The equatorward flow of freshwater is limited up to 18°N due to divergence between the EICC and a weaker WBC. In AS, the surface waters experience huge evaporation due to strong winds thereby increasing the salinity. Higher evaporation over precipitation and vertical mixing with subsurface waters make sure that AS salinity remains well above 35 psu (Kumar and Prasad, 1999). The equatorward-flowing WICC brings relatively salty water into BoB via the
southwest monsoon current (Vinayachandran et al., 2013). By ON season, EICC flows to conform to the southeastern coast of India. During DJF, we observe that the relatively fresher water flowing from BoB influences the salinity variations in the southeastern AS.

Similar to SST, the model captures the SSS spatial variations and seasonality well and conforms with the observed SSS. The
salinity difference between AS and BoB is substantial in all seasons, which is also simulated well by the model. In DJF, model SSS is saltier in the eastern BoB, as compared to observations (Fig. 3p, t). This bias can be attributed to the slight overestimation in the eastern boundary forcing data which influences the simulated SSS of this region. Statistically, the model SSS shows a high spatial Pearson's correlation with both SMOS and gridded Argo SSS in all seasons. We have also statistically validated the simulated subsurface temperature and salinity with RAMA buoys and gridded Argo data to a depth of 500m and 100m
respectively at various locations. This validation is included in the supplementary material (Text S1 and S2).





**Figure 4. Seasonal climatology of surface currents overlaid with vectors (in m/s) from (a-d) model, (e-h) GlobCurrent and (i-l) gridded INCOIS-HYCOM and bias of model with (m-p) GlobCurrent and (q-t) INCOIS-HYCOM. R values represent Pearson's correlation coefficients.**

We validated the model currents with GlobCurrent observations and compared them with INC-HYC simulated currents, averaged over the upper 15 meters. The model-simulated currents show consistent agreement with both the observations and the INC-HYC model. The model also captures the transition in coastal currents like WBC and EICC well. One interesting



thing to note is the eastward diversion of WBC along 18°N (Gangopadhyay et al., 2013), which is seen in GlobCurrent (Fig. 4e). However, in both the model and INC-HYC it flows further poleward. This phenomenon is also evident in the bias (Fig. 4m, q). During JJAS, equatorward flowing WICC brings waters from AS near the southern Sri Lankan coast, where the Sri Lankan Dome presides. During the same period, EICC forms over the northern BoB, which the model simulates following the other datasets. By ON, the EICC dominates the eastern coast of India and brings relatively fresher water into the southeastern AS, causing temperature inversions and barrier layer formation along this region (Mathew et al., 2018). During DJF, the model simulates the anticyclonic circulation of Laccadive High (LH) in the same region. In the AS, the WICC reverses its direction and flows poleward. We computed the bias of the current magnitude of our model with GlobCurrent and INC-HYC. The bias is well within limits and model surface currents spatially correlate well with both datasets (Fig. 4m-t).

To validate the model subsurface currents, we computed the alongshore currents at various locations and compared them with GlobCurrent over the upper 15 meters (Fig. 5). The stick plots of surface alongshore current at given locations (Figure 5 a-j), represent the current direction and magnitude as a function of time. We selected two locations on the west coast (Mumbai and Goa), two on the east coast (Cuddalore and Gopalpur), and one on the southern coast (Kanyakumari) of India to represent their respective alongshore characteristics (refer to Fig. 1).







**Figure 5. Stick plots of surface alongshore current of (a-e) model and (f-j) GlobCurrent and vertical alongshore current of (k-o) model and (p-t) GlobCurrent up to 15 meters at various slope locations along the AS and BoB.**

The latitude-longitude and angle of rotation for these locations were obtained from Amol et al. (2014) for the western and southern coasts and from Mukherjee et al. (2014) for the eastern coast. Sign convention for the alongshore current is defined as positive values for poleward flow and negative values for equatorward flow (Mukherjee et al., 2014). However, at Kanyakumari, the angle of rotation is 90° which makes the flow completely zonal. Here, positive values indicate eastward flow whereas negative values indicate westward flow. The alongshore currents along the west coast are weaker than the east

coast. This can be attributed to the coastal topography of the Indian peninsula in which the eastern coast is parallel to the seasonal winds and the western coast is mostly normal. This weakens the alongshore wind along the western coast and hence the currents (Amol et al., 2014). This is captured well by the simulation. However, the model simulates the seasonality better along the southern and eastern coasts as compared to the western coast. Model alongshore currents at Mumbai and Goa locations observe a lag in the transition of current from a poleward into an equatorward flow during February-April. We



observe the poleward flowing WBC during February-May and the equatorward flowing EICC during October-January at
Cuddalore and Gopalpur in both model and observations. At Kanyakumari, a strong south-eastward flow during JJAS brings
saltier water from AS to BoB, whereas the westward flow during October-January brings fresher water from BoB to AS.
Overall, the model performs well in simulating the seasonal variations of surface as well as subsurface alongshore currents.

**3.2 Alongshore Volume transport (AVT) and Cross-shore Volume transport (CVT)**

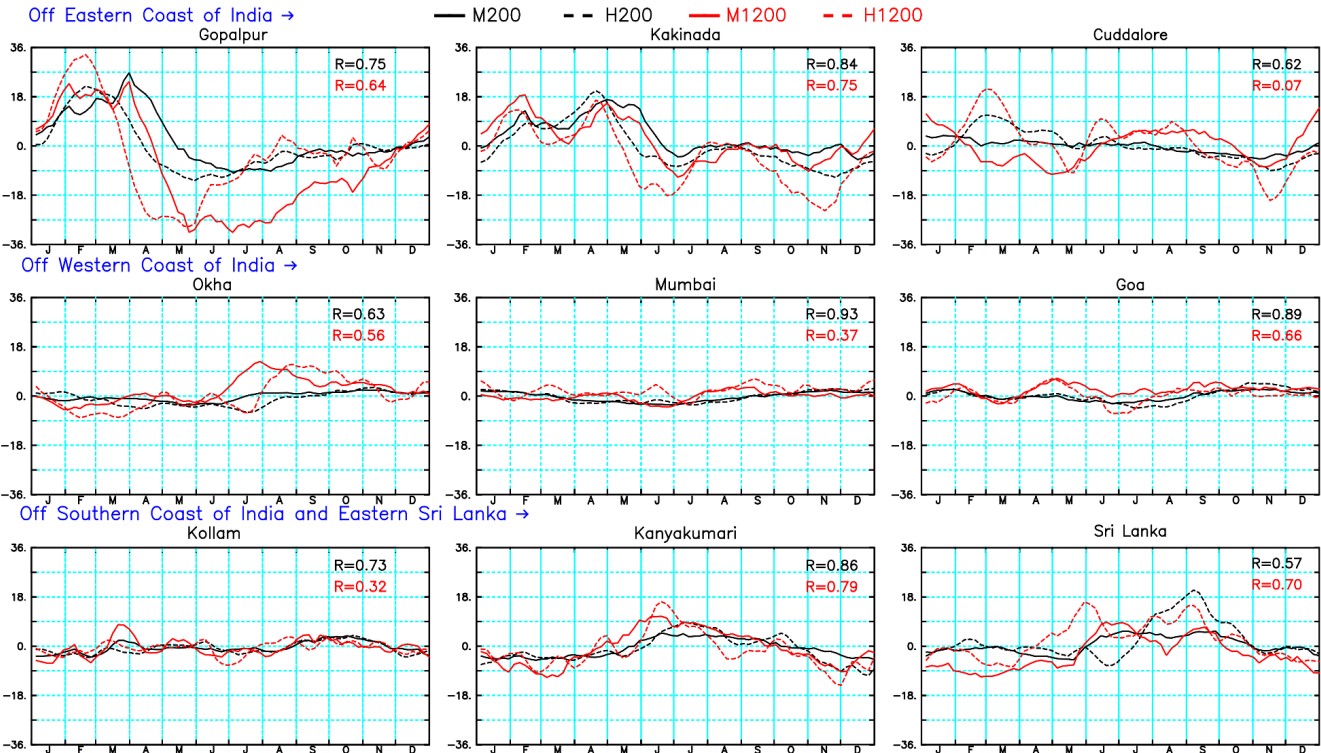

**Figure 6. AVT (in Sv) of the model (solid lines: M200 / M1200) and INC-HYC (dashed lines: H200 / H1200). R represents the Pearson's correlation coefficient between the model and INC-HYC. Articles in black colour indicate transport integrated over 200 m depth and red colour indicates transport integrated over 1200 m depth.**

To analyse the volume transports, we selected nine transects and computed the AVT and CVT. The three-degree width of the
transects was selected to incorporate the WICC and EICC, which typically flow conforming to the 1000 m isobath (Shetye et
al., 1991a; Amol et al., 2014; Mukherjee et al., 2014) (refer Fig. 1). We considered three transects each along the eastern
(Gopalpur, Kakinada and Cuddalore) and western (Okha, Mumbai and Goa) coasts. Two transects are located along the
southern coast of India (Kollam and Kanyakumari) and one transect is located along the eastern coast of Sri Lanka. The AVT

were plotted as a function over time to observe their seasonal and intraseasonal variability. Positive values indicate poleward
transport, whereas negative values indicate equatorward transport. For the Kanyakumari and Sri Lanka transects, positive
values indicate eastward (northward) flow, while negative values indicate westward (southward) flow. We assume the AVT





to be purely meridional flow at the eastern coast of Sri Lanka. Black lines represent the transport computed over the upper 200m (M200 or H200) to understand the seasonal variability over the upper ocean, whereas red lines represent transport over the upper 1200m (M1200 or H1200) to estimate the influence of undercurrents and their contribution towards the seasonality of subsurface transport.

Along the eastern coast, the seasonal cycle dominates the transport of EICC (Mukherjee et al., 2014; Mukhopadhyay et al., 2020). However, we observe some intraseasonal variability during pre-summer monsoon (February-April). The flow of the transport reduces from north to south. The equatorward (negative) transport (Fig. 6, M1200) indicates a strong subsurface undercurrent during February-April. It is interesting to note that surface alongshore transport is highest at Gopalpur and lowest at Mumbai, both being at almost the same latitude. This is because the transport along the eastern coast is largely seasonal whereas on the other coasts, it is modulated mostly by intraseasonal oscillations. These intraseasonal features are also prominent in the subsurface transport of the western and southern coasts. In the west coast locations along the WICC, we observe weaker seasonal variations. Over Okha, we see a strong subsurface poleward transport (Fig. 6, M1200/H1200) peaking during the summer monsoon, which is not observed in the surface layers. In the southern part, the AVT at Kollam uniformly deviates around its mean signal throughout the year, which has also been noted by the current pattern in Chaudhuri et al. (2020). The transport at Kanyakumari is dominated by the eastward southwest monsoon current during the summer monsoon and the westward northeast monsoon current during the winter. Along the coast of Sri Lanka, our estimates of AVT agree with Anutaliya et al. (2017). Two poleward peaks are observed in this region, first during June-July and the second during September-October.

We compared the model-simulated transport values with the ones from INC-HYC and computed correlation coefficients (Fig. 6). Statistically, we observe a strong positive correlation between the transport from both models and INC-HYC. Our estimates are also consistent with the findings from Shetye et al. (1991b, 1996). The model and INC-HYC computed monthly values of AVT are presented in the supplementary material, Table S1. The heat component of transport i.e., AHT has also been computed to provide a complete picture. The seasonality of computed AHT is similar to that of AVT. There is no significant difference between surface and subsurface heat transport, which shows that substantial transport of heat only occurs at the upper layers of the ocean. (refer to supplementary material, Fig. S3).



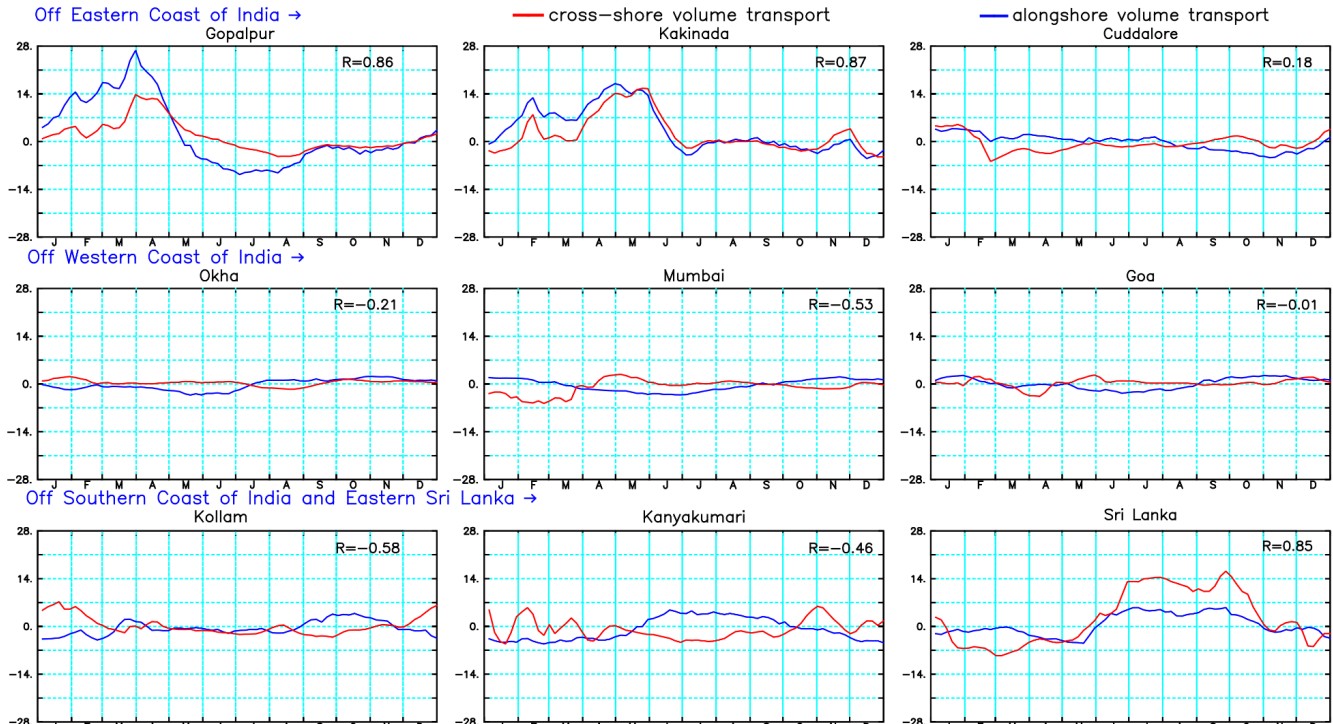

**Figure 7. Comparison of model computed AVT (solid dark blue line) and CVT (solid red line), integrated over 200 m depth.**

We also analysed the relationship between the upper 200m model AVT and corresponding cross-shore transport i.e., CVT at these locations. The sign convention for cross-shore transport has been referred from Mukherjee et al. (2014), where offshore (upwelling favourable) transport is positive and inshore (downwelling favourable) transport is negative. Similar to the alongshore component, the seasonal characteristics of CVT over the upper 200m on the eastern coast are stronger than the other locations. On the other coasts, intraseasonal variability is more prominent. An interesting observation is that the CVT along the east coast exhibits synchronous phasing with the AVT, in contrast to the west coast where it shows an out-of-phase relationship. Further elucidation of this alongshore and cross-shore relationship for both coasts is detailed below:

For the western coast:

(a)     If the alongshore component is equatorward (-ve), the cross-shore component is offshore (+ve).

(b)     If the alongshore component is poleward (+ve), the cross-shore component is inshore (-ve).

For the eastern coast:

(a)     If the alongshore component is equatorward (-ve), the cross-shore component is inshore (-ve).

(b)     If the alongshore component is poleward (+ve), the cross-shore component is offshore (+ve).

This finding is also supported by the correlation values between AVT and CVT (Fig. 7). On the eastern coast, we observe a strong positive correlation between CVT and AVT at Gopalpur and Kakinada. Like AVT, the CVT also grows weaker in the southern part of this coast. At Cuddalore, the CVT is inshore during MAM season when the AVT is poleward. A similar



observation about the cross-shore current patterns is also reported by Mukhopadhyay et al. (2020). Along the west coast, the
observed CVT is strongest at Kollam as compared to Okha, Mumbai, and Goa (Amol et al., 2014; Chaudhuri et al., 2020) as
also observed in Fig. 7. Another important pattern is observed during the winter in this coast. The AVT is equatorward whereas
the CVT is offshore, with a peak value of ~7 Sv by January. Also, the magnitude of CVT is stronger than AVT at Mumbai,
Goa, and Kollam whereas it is weaker at Okha. This is clear from the strong negative correlation at Kollam (r = -0.58) and
Mumbai (r = -0.53) and a relatively weaker negative correlation at Okha (r = -0.21). No correlation is observed at Goa. On the
southern coast at Kanyakumari, the sign convention of CVT is positive (negative) southward (northward). The CVT is highly
intraseasonal here, as compared to AVT. This can be attributed to the influence of equatorial forcing over this region. Both the
transports are negatively correlated to each other (r = -0.46). At Sri Lanka, the cross-shore transport sign convention is positive
(negative) eastward (westward). Both the transports are seasonally analogous and positively correlate (r = 0.85) with each
other. The poleward (equatorward) alongshore component corresponds to an outward (inward) cross-shore component in this
region. As in the case of AVT, we observe large intraseasonal variability in CVT as well. Amol et al. (2014) identified the
reason to be eddy-induced smaller circulations which influence the flow of currents and thereby the transports.

## 3.3 Alongshore Freshwater Transports (AFT)

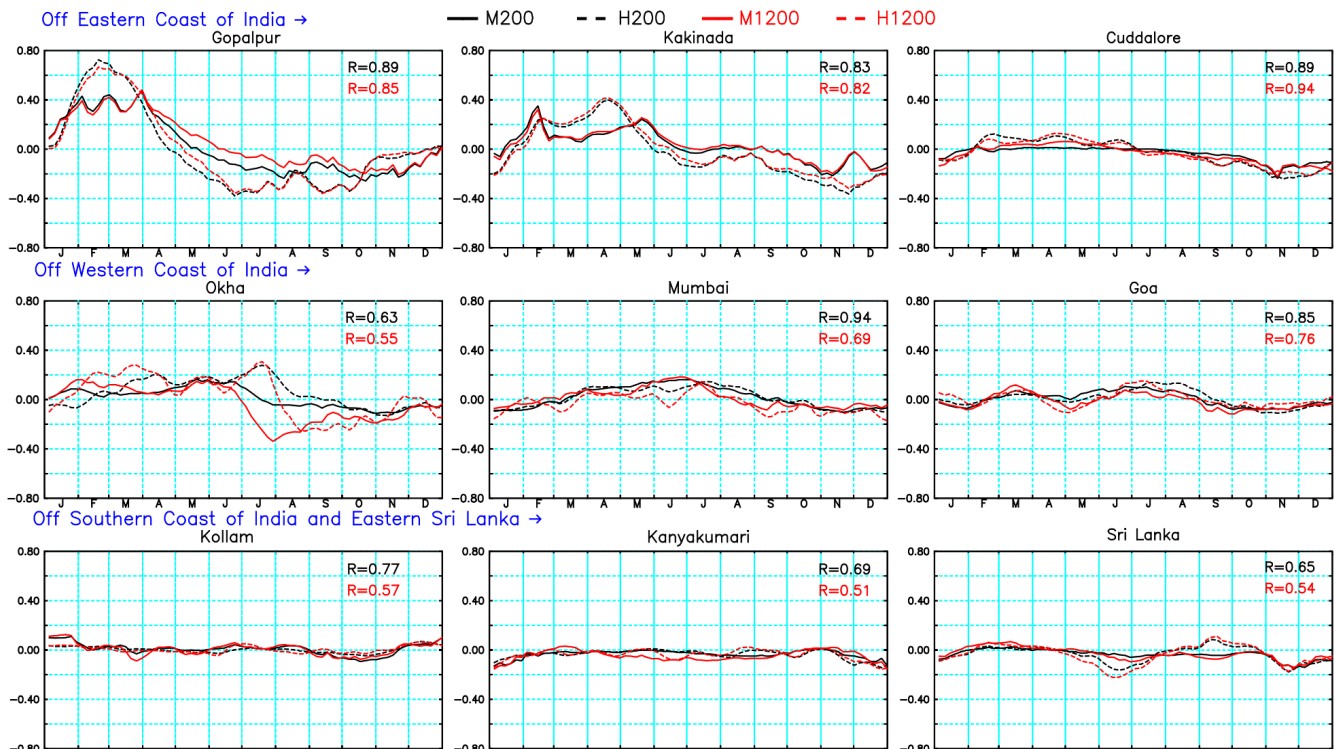





**Figure 8.** AFT (in Sv) of the model (solid lines: M200 / M1200) and INCOIS-HYCOM (dashed lines: H200 / H1200). R represents the Pearson's correlation coefficient between the model and INC-HYC. Articles in black colour indicate transport integrated over 200 m depth and red colour indicates transport integrated over 1200 m depth.

Along the Indian coasts, the analysis of freshwater transports is crucial. This importance arises from the significant contributions of precipitation and discharge of glacial meltwater, both of which are integral components of coastal dynamics.

For this, we computed AFT by taking the double integral of freshwater fraction and alongshore velocity component in length and depth at the same locations as the AVT and CVT (Fig. 8). We computed the AFT in Sverdrup (volume) instead of kg/s (mass) to maintain uniformity. We observe some interesting patterns in the AFT variability. First, the direction of seasonal variations of AVT and AFT contradict each other on the western coast. However, they are similar on the eastern coast. This can be attributed to the role of salinity in both basins. Second, we note that the magnitude of AVT is about two orders higher

than that of AFT. Third, the transports computed within 200m and 1200m depths are almost coincident with each other. This shows that the involvement of subsurface transport in freshwater exchange is not very significant. However, at Okha, we observe a considerably strong equatorward transport during JJAS.

Comparing the coasts, the AFT on the eastern coast is greater than on the western coast, similar to AVT. During February-

April, the WBC transports freshwater poleward along the eastern coast. As the salinity in BoB is always lesser than the reference ($S_{ref}$) and the alongshore current component is positive (refer to section 2.3), the directions of AVT and AFT are the same as each other in this period. As the summer monsoon arrives, we see an equatorward flow of freshwater driven by EICC due to a negative alongshore current component. This equatorward flow is prominent during the JJAS season at Gopalpur. It reaches Kakinada during September- October and ultimately reaches Cuddalore by November. The AFT over the western

coast is very weak, especially near Kollam and Kanyakumari. This happens majorly because salinity does not fluctuate much in this region and remains close to the $S_{ref}$ value. We observe a weak positive AFT at Mumbai and Goa during the summer monsoon. This can be attributed to the high precipitation along these regions in addition to the river runoff which is discharged into the coastal AS (Behara et al., 2019). Also at Kanyakumari, we observe a weak seasonality. A dip of ~ 0.15 Sv is observed during DJF, which shows that the freshwater is transported westward from southwestern BoB into southeastern AS by the

northeast monsoon current. Along the coast of Sri Lanka, we observe a dip in AFT during November-December. This is because the equatorward-flowing EICC brings freshwater into this region. Our estimates of AFT at Sri Lanka agree with the findings of Rainville et al. (2022). Also, the model AFT estimates at all locations conform well with the AFT computed from the INC-HYC model (Fig. 8). Similar to AVT, the surface correlation between the two models is better than the subsurface correlation. The monthly computed values of AFT are presented in the supplementary material, Table S2.


**3.4 Spatial variability of Net Volume Transport (NVT) and Net Freshwater Transport (NFT)**





**Figure 9. Seasonal spatial variability of NVT and NFT of model and INC-HYC. Positive values (red) indicate northward/eastward flow whereas negative values (blue) indicate southward/westward flow.**

To understand the seasonal spatial transport over the domain, we computed NVT and NFT at each grid point integrated over

the upper 100 m (Fig. 9). When analysing transports at transects along the coasts, the alongshore and cross-shore components

become pertinent. However, for a grid-wise computation in the domain, the net component involving the zonal and meridional

information is a more robust and meaningful metric. As observed in previous sections, the transport along coastal BoB is

stronger as compared to AS. In MAM, both the model and INC-HYC simulate a strong poleward NVT due to WBC extending

from 10°N-20°N. Simultaneously, a weaker westward flow (2.5 Sv to 5 Sv) streams along the southern coast of Sri Lanka.



During JJAS, a pair of cyclonic eddies situated over the northwestern BoB is observed. These eddies are active during June-July, but by August-September, they get disrupted due to an equatorward transport initiated by the EICC. During this period, we also observed the Sri Lankan Dome along the eastern coast of Sri Lanka. It forms during early June and strengthens into a cyclonic circulation (Cullen and Shroyer, 2019). The NVT in this region exceeds 10 Sv during JJAS and weakens by ON as it
moves northward. In ON, the EICC in BoB has a discontinuous flow (Durand et al., 2009) and shifts equatorward along the southwest BoB. By DJF, it is observed that the equatorward flow of EICC gets further carried westward by the northeast monsoon current into southeastern AS but remains confined between 4°N-6°N. This is simulated well by both models and is also reported by Zhang and Du (2012). Overall, model NVT over the upper 100 m is stronger than INC-HYC.

The seasonal NFT pattern is similar to that of NVT. However, its value is an order less than NVT. We have quantified the NFT as a percentage of NVT over different seasons to understand how much freshwater gets advected by WBC and EICC along the eastern coast of India (refer to supplementary material, Table S3). During MAM, the NFT of WBC is 2.096 % and 3.617 % of the NVT from the model and INC-HYC respectively. Low saline water of 0.1-0.3 Sv is advected poleward in this period. In the JJAS season, the equatorward EICC transports around 0.2-0.4 Sv of freshwater. It is bounded between the head
and northwestern BoB and transports most freshwater (6.03 % model and 8.39 % INC-HYC). Also, the major Indian rivers have the largest riverine discharge during this season (Dai and Trenberth, 2002). By ON, the equatorward NFT along the eastern coast of India reaches southwest BoB and transport exceeds 0.4 Sv at few locations. In DJF, the freshwater is further transported westward into southeastern AS. The flow seems to be strongest along the southeastern coast of India and the southern coast of Sri Lanka (> 0.3 Sv). As the water propagates from BoB into southeastern AS, its freshness decreases slowly
due to continuous mixing with saltier waters (refer to supplementary material, Table S3). Our results along the coast of Sri Lanka agree with Rainville et al. (2022).

**3.5 Net Heat Transport (NHT) and Net Heat Flux**





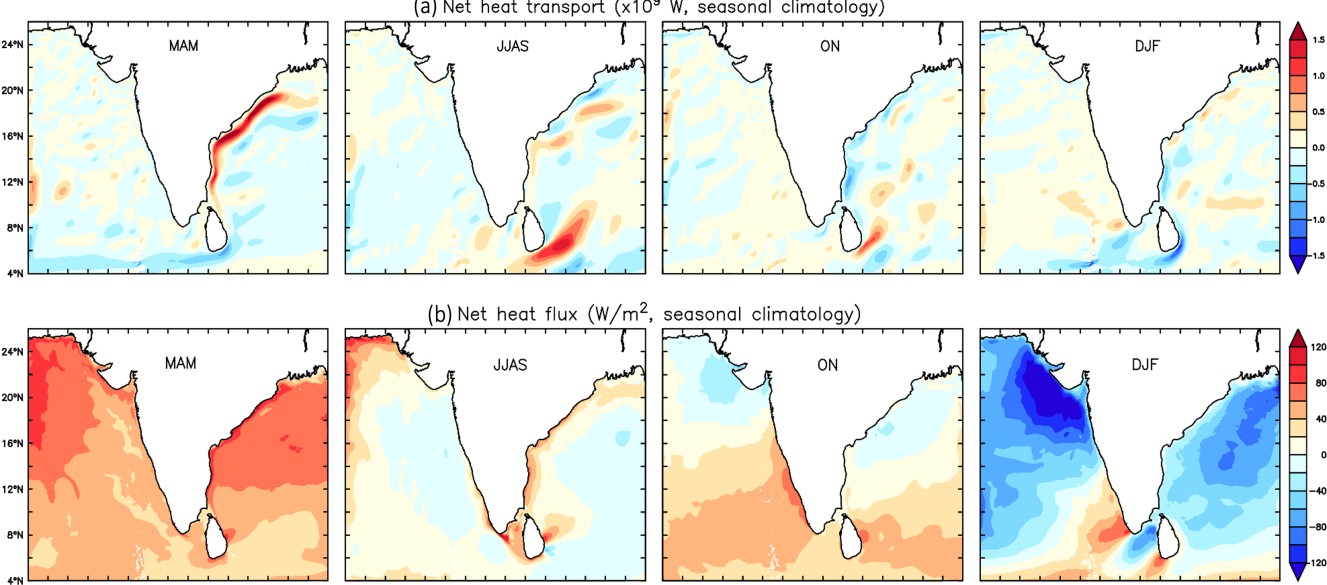

**Figure 10. (a) seasonal NHT (GW, 1GW= 1x10⁹ W) integrated over the upper 100m. For the upper panel, (a) colour red indicates northward/eastward flow whereas blue indicates southward/westward flow, (b) seasonal surface net heat flux (W/m²). For the lower panel, (b) colour red represents heat gain and blue represents heat loss.**

Studying the NHT in addition to volume transports provides a more comprehensive view, as heat transport is an important link

influencing the dynamics of water movement. We aimed to understand the thermal component of the transports and its spatio-temporal variability in the NIO. The spatial characteristics of NHT (Fig. 10a) are similar to that of NVT (Fig. 9). In MAM, the WBC carries the warmer water along a strong poleward transport. With the onset of June, the NHT is prominently high along the Sri Lankan Dome. This cyclonic eddy is strengthened by JJAS, and the warm water is transported northwards by ON. This is clear in the NHT signature of JJAS and ON. In the winter season, the heat transport is relatively low because of several

factors. The cooler and drier winds during winter and reduced solar radiation cool the temperature of the upper layers.

The characteristics of net heat flux are considerably different from that of the NHT. The heat flux is primarily governed by the atmospheric forcings and vertical mixing. During MAM, there is an increase in net shortwave radiation and the winds are weak which stalls the vertical mixing. This causes shoaling of the mixed layer depth, increases stratification and traps heat in the

upper surface layers in almost the entire domain. In JJAS, we observe the cooling of surface waters due to rain-bearing clouds, which block incoming shortwave radiation (Das et al., 2016). Besides this, large evaporation results in the release of latent heat which increases cooling majorly in the open waters (Das et al., 2016). The western AS is saturated due to an increase in the specific humidity of air (Pinker et al., 2020). This further prevents evaporation leading to a net heat gain in JJAS. At the same time, the waters along the eastern coast of India in BoB gain heat owing to stratification. By ON, net shortwave radiation



decreases and the wind direction reverses. The northeasterly cold dry winds cool the surface waters of northern AS and BoB,
which triggers convective mixing. This causes heat loss in the form of latent heat which is further intensified by DJF in these
parts of the basin (Kumar and Prasad, 1996). We observe similar cooling near the southern tip of the mainland owing to latent
heat release due to strong winds (Luis and Kawamura, 2000). The southeastern AS region has high heat flux throughout the
ON, DJF and MAM seasons. This is attributed to the export of freshwater from southwestern BoB and a downwelling Rossby

wave which creates a thick barrier layer and prohibits vertical mixing, thus capping heat in the upper layers (Mathew et al.,
2018).

### 3.6 Evaluation of meridional heat transport

To understand the meridional heat distribution and ventilation in the NIO basin by currents, we computed the MHT in AS and

BoB basins for the model and INC-HYC. We zonally integrated MHT at each latitude, over the upper 100m. The MHT is
plotted as a function over time (Fig. 11). We observe that the pattern of MHT is similar in the model and INC-HYC. However,
the magnitude of MHT is higher in INC-HYC than in the model.

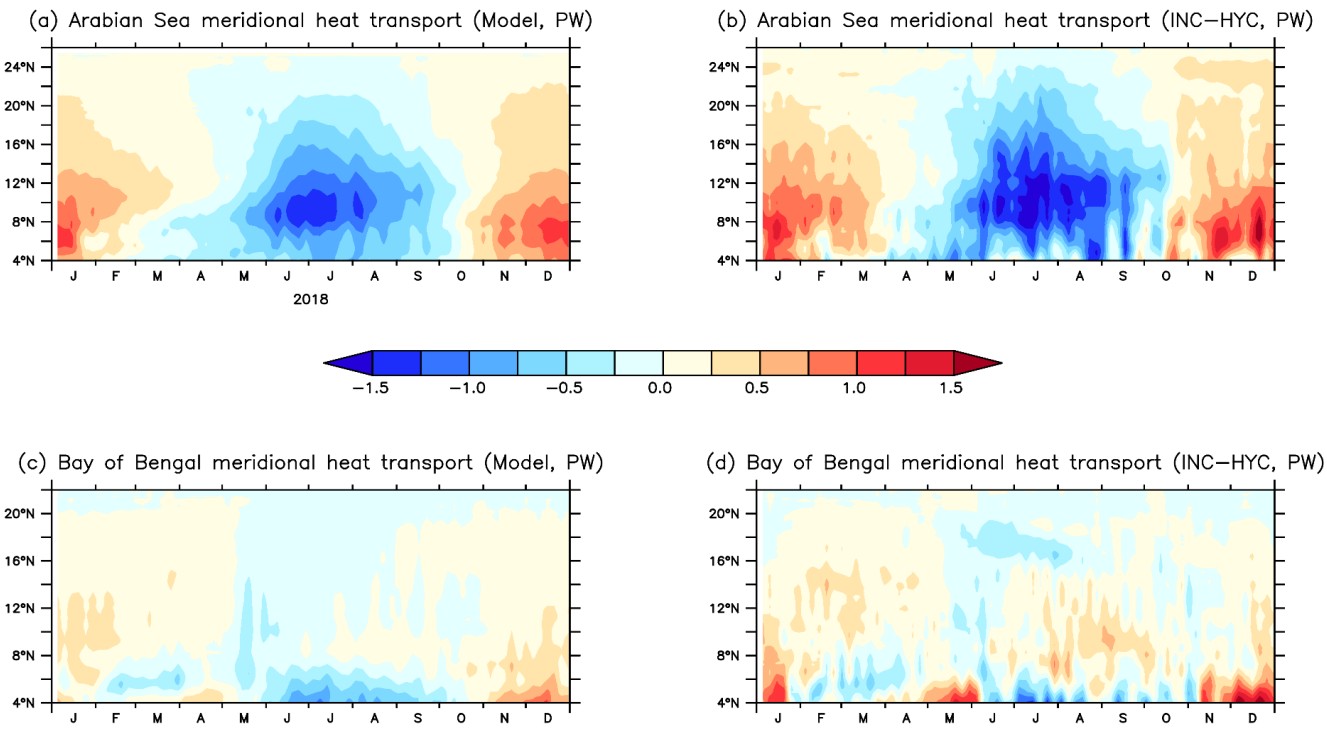

**Figure 11. Meridional heat transport (MHT, in PW) computed over the upper 100m using the model and INC-HYC for AS and**

**BoB. Red colour indicates northward transport and blue colour indicates southward transport.**

The MHT pattern of AS is considerably different from that of BoB. The AS MHT has a prominent seasonal signature with a
southward transport exceeding 1.25 PW during the JJAS and a northward transport (> 1 PW) during the DJF (Fig. 11 a, b). In
other words, the heat flows out from the AS into the equatorial Indian Ocean during the summer monsoon and the heat flows



into AS in the winter (Garternicht and Schott, 1997). This reversal of MHT is attributed to the reversal of winds and meridional Ekman transports, in addition to the contribution from vertical thermal wind shear as reported by various studies (Wacongne and Pacanowski, 1996; Lee and Marotzke, 1998; Schott and McCreary, 2001). The latitudinal band between 6°N-12°N sees the strongest positive and negative signals of MHT. The propagation of an annual bi-mode westward Rossby wave triggered along the western coast of India is a reason behind this influence on the strength of MHT in this band (Brandt et al., 2002). The reversal in the direction of MHT occurs during the MAM and ON seasons. It is also a known fact that the weakening of the summer monsoon can impact the Ekman transport, thereby weakening southward MHT and further warming of AS (Swapna et al., 2017; Pratik et al., 2019). BoB MHT is weaker than AS (Fig. 11 c, d). It is attributed to higher stratification and weaker winds (Mahadevan et al., 2016; Shenoi, 2002). Weak winds cause slower circulation and poor vertical mixing causing a reduction in MHT (Mallick et al., 2020). Also, it is an active eddy region (Chen et al., 2012; Cheng et al., 2018). Overall, the MHT does not exceed ± 0.75 PW in this basin. We observe a weak southward MHT (~ 0.25 PW) during the JJAS and a weak northward MHT (~ 0.5 PW) during the DJF and MAM seasons. The values agree with Shi et al., (2002). Along 6°N latitude, ~ 0.5 PW of southward MHT exists between February and April. The lower latitudinal band between 4°N-6°N shows a comparatively stronger intra-seasonal modulation of MHT. We find small bursts of northward MHT between November-January and April-May, and southward MHT during JJAS. This is influenced by the transport from equatorial IO. The contribution of eddy-associated transport is less significant since their values are an order less than basin-scale values (Lin et al., 2019). Overall, we understand that NIO becomes a heat source (sink) during the summer (winter) season. The heat stored in the upper waters gets flushed out due to an overall southward transport during March-September whereas, water from the equatorial Indian Ocean is brought in by the northward transport during November-February.

**4 Summary**

In summary, our high-resolution climatological model setup, as demonstrated in this study, provides a robust foundation for advancing our understanding of volume, freshwater and heat transport dynamics in the North Indian Ocean. The eddy-resolving capability and finer bathymetry in high-resolution modelling improve the overall estimates of exchange along and in the coastal waters. The model accurately captures significant patterns. Major patterns observed are that the Alongshore Volume Transport (AVT) is stronger on the eastern coast and is highly seasonal, as compared to the western coast, where it is influenced by large intraseasonal oscillations. We observe an inverse relationship between AVT and Cross-shore Volume Transport (CVT) on the western coast and a direct relationship on the eastern coast. Seasonal variations between AVT and Alongshore Freshwater Transport along the western coast also present such a contradiction, while on the eastern coast, they display in-phase behaviour. This depends majorly on the intricate coastal dynamics influenced by salinity variations in AS and BoB. Spatially, our analysis indicates a similar seasonal pattern between Net Freshwater Transport and Net Volume Transport, with the freshwater transport values notably lower than the total transport. Out of the NVT, freshwater accounts for the maximum during the JJAS season (6.03 %) primarily confined to the head BoB and northeastern coast of India. This is followed by the ON season (4.85 %) when it flows along the entire eastern coast. The relation between NHT and net heat flux illustrates the role of coastal currents



and equatorial forcing in dissipating heat within the coastal waters. Another important finding is that the Meridional Heat Transport (MHT) is stronger in the AS compared to the BoB. The MHT plays a crucial role in flushing heat out during summer monsoon seasons while bringing equatorial waters in during the winter season.

The model setup successfully simulates these vital climatological patterns, emphasizing the significance of high-resolution modelling in understanding the complex ocean dynamics in this region. This work serves as a comprehensive study of model setup intricacies, providing valuable insights into freshwater and heat distribution patterns both at the surface and in the depths of the basin. The established high-resolution physical model, coupled with biogeochemical modules, can be used as a promising tool for further investigations into the biological processes and the transport of nutrients and carbon along the Indian coastal seas. Ultimately, this work contributes significantly to our understanding of oceanic dynamics, paving the way for more nuanced research into the productivity and ecological aspects of this region.

**Code and data availability**

GEBCO bathymetry data has been downloaded from https://www.gebco.net/data_and_products/gridded_bathymetry_data/. WOA18 ocean temperature and salinity are obtained from https://www.ncei.noaa.gov/access/world-ocean-atlas-2018/. SODA 3.12.2 data can be accessed from https://www2.atmos.umd.edu/%7Eocean/index_files/soda3.12.2_mn_download_b.htm. The ERA5 data on a single level is obtained at https://cds.climate.copernicus.eu/cdsapp#!/dataset/reanalysis-era5-single-levels?tab=form. ERA5 specific humidity has been obtained from https://cds.climate.copernicus.eu/cdsapp#!/dataset/reanalysis-era5-pressure-levels?tab=form. GPM precipitation data can be downloaded from http://apdrc.soest.hawaii.edu/datadoc/gpm_imerg_mon.php. MUR SST is downloaded from https://podaac.jpl.nasa.gov/dataset/MUR-JPL-L4-GLOB-v4.1. SMAP SSS is downloaded from https://podaac.jpl.nasa.gov/dataset/SMAP_JPL_L3_SSS_CAP_MONTHLY_V5. GHRSST can be downloaded from https://podaac.jpl.nasa.gov/dataset/OSTIA-UKMO-L4-GLOB-v2.0. SMOS SSS is downloaded from https://www.seanoe.org/data/00417/52804/. CORA data is downloaded from https://www.seanoe.org/data/00351/46219/. GlobCurrent currents data is available at https://data.marine.copernicus.eu/product/MULTIOBS_GLO_PHY_REP_015_004/services. MATLAB and Pyferret software were used for analysis and data visualization. The HYCOM model-simulated data used in this manuscript can be made available at https://incois.gov.in/portal/datainfo/drform.jsp.

**Author Contributions**

Conceptualization, K.M. and A.D.R.; methodology, K.M. and A.D.R.; software, K.M.; validation, K.M.; formal analysis, K.M.; investigation, K.M. and S.J.; resources, A.D.R.; data curation, S.J.; writing—original draft preparation, K.M.; writing—review and editing, K.M., A.D.R. and S.J.; visualization, K.M.; supervision, A.D.R. All authors have read and agreed to the published version of the manuscript.





**Competing interests**

The authors declare that they have no conflict of interest.


**Acknowledgements**

The first author acknowledges IIT Delhi for PhD fellowship support.

**Funding support**

This research received no external funding.

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
