# Peer review of "High-resolution numerical modelling of seasonal volume, freshwater, and heat transport along the Indian coast"

_EGUsphere, 2023_

## Referee Comment (RC1)

Review of MS by Madkaiker et al

**High-resolution numerical modelling of seasonal volume, freshwater, and heat transport along the Indian coast**

The MS uses advanced research techniques to study transport processes along the Indian coast. The main numerical model is MITgcm model that uses high for oceanic conditions spatial resolution of 1/20° (about 3 nautical miles, ca 5 km). The model is forced with climatological initial and boundary conditions. The model was run for five-year hindcast period (2012-2016) with daily output fields. To reduce the spin-up time and computational expenditure, the model is initialized with a warm start. For this, climatological zonal and meridional currents are prescribed from the Simple Ocean Data Assimilation (SODA) 3.12.2 dataset. The model results were compared with another model, the HYCOM model which was routinely run by Indian national authority with 1/16° resolution. The developed model was validated with a number of observational products: remotely sensed GHRSST and SMOS for sea surface temperature and salinity, gridded reanalysis CORA v5.2, and GlobCurrent zonal and meridional surface currents.

The research has been conducted according to contemporary standards. The model has high resolution for the ocean basins, the results of such a setup are verified. With such a start, the analysis methods are not up-to-date and are mainly adopted from ocean-scale oceanography. The results on volume, freshwater and heat transports are of interest, but it is not clear how much they go beyond the existing knowledge presented in the introduction and other publications. The introduction states: "The broad goal of this study was to set up the MITgcm model over our domain and estimate surface and subsurface volume, freshwater and heat transports in the basin." The model was successfully set up, but this goal and the results seem too narrow for a good scientific publication. Therefore, a revision of the MS is proposed.

**A. Transports due to mesoscale eddies**

The model is eddy-resolving. Therefore, it is useful to get knowledge of eddy-driven fluxes, e.g. by separating the freshwater and heat transports into the mean and eddy-driven parts. There are many such studies published, for example Ding et al. (2021). The updated study could also look at freshwater plume and fronts in the northern Bay of Bengal. At the same time, comparison with routine HYCOM model should be significantly condensed, both in the figures and in the text.

Ding, R., Xuan, J., Zhang, T., Zhou, L., Zhou, F., Meng, Q. and Kang, I.S., 2021. Eddy-induced heat transport in the south China Sea. Journal of Physical Oceanography, 51(7), pp.2329-2349.

**B. Reference for freshwater transport**

Large rivers transport freshwater to the sea as shown in Fig. 1. Impact of river discharge is evident in Fig. 3. In the Bay of Bengal salinities below 31 g kg$^{-1}$ extend to the large sea area from the major rivers in the north, like Brahmaputra and Ganga.

The MS does not provide information where the salinity reference value 34.83 psu comes from and what are the justifications behind. The cited study by Stammer et al. (2003) does not define fixed reference salinity. From the references in the latter paper, Wijffels et al. (1992) present overall frame for flux calculation through the defined cross-sections. They also separate the transports due to the mean velocity and salinity values over the sections, and the component due to their variations within the section (see also comment A). Some key to the salinity reference value can be found in Rainville et al. (2022) who indicate that in an earlier study that 34.83 g kg$^{-1}$ has been found as a mean salinity of the (open) Indian Ocean. The present MS gives 34.83 psu that is not correct adoption from the mentioned paper, since 34.83 g kg$^{-1}$ = 34.67 psu.

The issue of reference salinity should be reconsidered.

**C. Angles of the coast**

There is strong misunderstanding in lines 167-168 and 173-174 that establish different equations (1)-(2) and (3)-(4) for calculation of alongshore and cross-shore current components at different coasts. In mathematics, angle definitions and equations for projections of the vectors are uniform, but the angle values are different. Strict mathematical approach should be used for defining the vector components.

One approach could be following. In any point of the "coastal" curve surrounding the land anticlockwise, we can draw the tangent vector to the curve at that point. The normal vector is then at right angles to the curve, so it is also at right angles (perpendicular) to the tangent. If extended trigonometry is needed, then it could be moved to an appendix.

**D. Structure of the MS**

The goals in the Introduction should be improved and clearly spelled out. There should be more oceanography than new setup and validation of the model. Discussion could be moved into the separate chapter, presently it is not easy to read. Summary (perhaps Conclusions) could be cleared from general statements, like "Ultimately, this work contributes significantly to our understanding of oceanic dynamics, paving the way for more nuanced research into the productivity and ecological aspects of this region."

There are also some technical comments.

1) Line 72: Abbreviation NC-HYC appears without explanation. For sequential reading it does not help that it is explained later in Line 124.

2) Line 78: Axis of Fig. 1 are not defined. Also, for Figs. 2 to 11.

3) Lines 171-172: References to Amol et al. (2014) and Mukherjee et al. (2014), are they needed?

4) Line 176: Definition of Sverdrup has an error.

5) Line 203: Fig. 2 presents seasonal SST climatology from different data sets. How well is the climatology determined, is averaging done over the same periods and data density.

6) Line 258: Fig. 4 introduces INCOIS-HYCOM. What is this?

7) Line 264: Naming problem in "in both the **model** and **INC-HYC** it flows further poleward". The same problem in lines 298, 330, 371, 403, 409, 423, 465, 466, 469.

---

## Author Comment (AC1)

**Comment on egusphere-2023-3011 by Anonymous Referee #1**
**Title: High-resolution numerical modelling of seasonal volume, freshwater, and heat transport along the Indian coast**

**We thank the reviewer for the comments which have helped us improve this manuscript. Given below are detailed replies to each of their comments. The reviewer's comments are in black and our responses are in red. The text from the manuscript, added or modified, can be identified by blue in italics and quotation marks.**

The MS uses advanced research techniques to study transport processes along the Indian coast. The main numerical model is MITgcm model that uses high resolution for oceanic conditions with spatial resolution of 1/20° (about 3 nautical miles, ca 5 km). The model is forced with climatological initial and boundary conditions. The model was run for a five-year hindcast period (2012-2016) with daily output fields. To reduce the spin-up time and computational expenditure, the model is initialized with a warm start. For this, climatological zonal and meridional currents are prescribed from the Simple Ocean Data Assimilation (SODA) 3.12.2 dataset. The model results were compared with another model, the HYCOM model which was routinely run by Indian national authority with 1/16° resolution. The developed model was validated with a number of observational products: remotely sensed GHRSST and SMOS for sea surface temperature and salinity, gridded reanalysis CORA v5.2, and GlobCurrent zonal and meridional surface currents.

The research has been conducted according to contemporary standards. The model has high resolution for the ocean basins, and the results of such a setup are verified. With such a start, the analysis methods are not up-to-date and are mainly adopted from ocean-scale oceanography. The results on volume, freshwater and heat transports are of interest, but it is not clear how much they go beyond the existing knowledge presented in the introduction and other publications. The introduction states: "The broad goal of this study was to set up the MITgcm model over our domain and estimate surface and subsurface volume, freshwater and heat transports in the basin." The model was successfully set up, but this goal and the results seem too narrow for a good scientific publication. Therefore, a revision of the MS is proposed.

The authors thank the reviewer for the encouragement and suggestions to improve the quality of this manuscript. We would like to bring to the reviewer's attention that it is rather the HYCOM and not the MITgcm model that has been simulated for a five-year hindcast period (2012-2016). The mean of these five years was considered for comparing the transport results with the MITgcm model. In this study, the MITgcm model has been set up using climatological initial, boundary conditions and surface forcings. It was spun up for 5 years and the 6th year output was considered for analysis. We have clarified this in the revised manuscript to avoid such confusion for readers. We have also revised the equations used for the computation of alongshore transport

as per the suggestions of the reviewer and also included additional analysis. The broad goal statement has been revised:

*'To understand the exchanges of heat and freshwater by quantifying transports along the coastal pathways of the Arabian Sea and the Bay of Bengal using a robust MITgcm model set-up'.*

**A. Transports due to mesoscale eddies**
The model is eddy-resolving. Therefore, it is useful to get knowledge of eddy-driven fluxes, e.g. by separating the freshwater and heat transport into the mean and eddy-driven parts. There are many such studies published, for example Ding et al. (2021). The updated study could also look at freshwater plume and fronts in the northern Bay of Bengal. At the same time, comparison with routine HYCOM model should be significantly condensed, both in the figures and in the text.

Answer: We thank the reviewer for this additional insight. We have incorporated this suggestion in the revised version of the manuscript. We have identified certain mesoscale eddies and have computed their contribution to transporting heat and freshwater, as suggested by the reviewer. These analyses have been added as a separate section in the revised manuscript. Following are the additional figures which we have added in the revised manuscript that represent the eddy-induced heat and freshwater transports in our domain.

(i) Eddy heat transport along the eastern coast of India

[Figure]

Figure 1: (a-c) Sea level anomaly overlaid with current vectors depicting the cyclonic and anticyclonic eddies simulated along the eastern coast of India. Vertical profile of ocean temperature, vertical velocity and MLD for Box 1 (d,e) and Box 2 (g,h) and their computed zonal, meridional and net eddy heat transports (f and i).

(ii) Eddy freshwater transport along the eastern coast of India

[Figure]

Figure 2: (a) SSS overlaid with current vectors depicting the cyclonic eddy simulated along the eastern coast of India. Vertical profile of ocean salinity, vertical velocity and MLD for Box 1 (b, c) and their computed zonal, meridional, and net eddy freshwater transports (d).

We would, however, like to clarify that analysing eddy-driven fluxes (such as Ding et al., 2021) requires interannual simulations. As the MITgcm model has been configured using climatological forcings, our results majorly suggest the climatological seasonal mean flow.

We have also condensed the usage of HYCOM in the revised manuscript. Also, we clarify that the HYCOM data is just to facilitate model-to-model comparison. Our entire analyses are based on MITgcm simulations. We have removed sections in the manuscript where the comparison with HYCOM was not required.

**B. Reference for freshwater transport**

Large rivers transport freshwater to the sea as shown in Fig. 1. Impact of river discharge is evident in Fig. 3. In the Bay of Bengal salinities below 31 gkg$^{-1}$ extend to the large sea area from the major rivers in the north, like Brahmaputra and Ganga.

The MS does not provide information where the salinity reference value 34.83 psu comes from and what are the justifications behind. The cited study by Stammer et al. (2003) does not define fixed reference salinity. From the references in the latter paper, Wijffels et al. (1992) present overall frame for flux calculation through the defined cross-sections. They also separate the transports due to the mean velocity and salinity values over the sections, and the component due to their variations within the section (see also comment A). Some key to the salinity reference

value can be found in Rainville et al. (2022) who indicate that in an earlier study that 34.83 gkg$^{-1}$ has been found as a mean salinity of the (open) Indian Ocean. The present MS gives 34.83 psu that is not correct adoption from the mentioned paper, since 34.83 gkg$^{-1}$= 34.67 psu. The issue of reference salinity should be reconsidered.

Answer: We thank the reviewer for pointing this out. We indeed verified that the reference salinity value in our study region is 34.67 psu. We also checked this against the reference mean value from SMOS and SMAP salinity data. The calculations pertaining to freshwater transport have been redone using 34.67 psu as the reference salinity. It has also been corrected in the revised manuscript.

**C. Angles of the coast**
There is strong misunderstanding in lines 167-168 and 173-174 that establish different equations (1)-(2) and (3)-(4) for calculation of alongshore and cross-shore current components at different coasts. In mathematics, angle definitions and equations for projections of the vectors are uniform, but the angle values are different. Strict mathematical approach should be used for defining the vector components. One approach could be following. In any point of the "coastal" curve surrounding the land anticlockwise, we can draw the tangent vector to the curve at that point. The normal vector is then at right angles to the curve, so it is also at right angles (perpendicular) to the tangent. If extended trigonometry is needed, then it could be moved to an appendix.

Answer: As per the reviewer's suggestion, the tangent vector approach was considered, and a normal vector of length 3 degrees each was drawn beginning from the coast. Further, the angle of rotation for calculation of the alongshore and cross-shore components was considered from the normal vector to the truth north, in an anticlockwise direction. We made sure that the equations for defining the alongshore and cross-shore components for both the west and east coasts remain the same. The updated transects have been corrected in the study area figure. The methodology has also been updated accordingly.

[Figure]

Figure 3: Method used to construct the transects along the coasts and to compute the angle of rotation. Here, 't' is the tangent vector, 'n' is the normal vector, 'N' is the true north, 'A' represents a typical transect and 'θ' is the angle of rotation with respect to the true north. The above figure represents how the angle is computed and is added to the appendix of the revised manuscript.

**D. Structure of the MS**

The goals in the Introduction should be improved and clearly spelled out. There should be more oceanography than new setup and validation of the model. Discussion could be moved into the separate chapter, presently it is not easy to read. Summary (perhaps Conclusions) could be cleared from general statements, like "Ultimately, this work contributes significantly to our understanding of oceanic dynamics, paving the way for more nuanced research into the productivity and ecological aspects of this region."

Answer: The broad goal in the introduction has been restructured for better clarity. Essentially, this work focuses on understanding the transports in the coastal waters along the Indian coastline using a robust climatological model set-up. We realized that the extensive validation confuses the readers concerning the actual goal of the manuscript. Hence we made the figures concise, moved a substantial part of the validation in the supplementary section and retained only the important parts. This is necessary to establish confidence in our model set-up. The summary and abstract have been revised and general statements have been removed from the revised manuscript.

There are also some technical comments.
1)Line 72: Abbreviation NC-HYC appears without explanation. For sequential reading it does not help that it is explained later in Line 124.
Answer: This has been addressed and line 72 has been rewritten as:

*'.... we conducted a model-to-model comparison of transports using another assimilated model data'.*

INC-HYC has also been explained at its first appearance in the manuscript.

2)Line 78: Axis of Fig. 1 are not defined. Also, for Figs. 2 to 11.
Answer: We have revised all figures accordingly and marked all their axes.

3)Lines 171-172: References to Amol et al. (2014) and Mukherjee et al. (2014), are they needed?
Answer: As the methodology to compute the angle of rotation has been revised, these citations are no longer required here and have been removed.

4)Line 176: Definition of Sverdrup has an error.
Answer: It was a typo and has been corrected.

5)Line 203: Fig.2 presents seasonal SST climatology from different data sets. How well is the climatology determined, is averaging done over the same periods and data density.
Answer: We used GHRSST (0.054°) and gridded Argo data (0.5°) to validate the model simulated SST. Both datasets were averaged over the same periods, with the monthly climatology computed for the last two decades, from 2000 to 2019. When comparing, the model SST was interpolated to the observation's spatial resolution.

6)Line 258: Fig. 4 introduces INCOIS-HYCOM. What is this?
Answer: This was a typo. We had written INCOIS-HYCOM instead of its abbreviation 'INC-HYC'. Model currents were compared with the assimilated HYCOM setup obtained from the INCOIS institute. It has been corrected in the figure.

7)Line 264: Naming problem in "in both the model and INC-HYC it flows further poleward". The same problem in lines 298, 330, 371, 403, 409, 423, 465, 466, 469.
Answer: This issue has been addressed. Our setup is denoted as 'MITgcm model' and the INCOIS-HYCOM setup is denoted as 'INC-HYC' to eliminate any confusion.

---

## Author Comment (AC2)

**Comment on egusphere-2023-3011 by Anonymous Referee #2**
**Title: High-resolution numerical modelling of seasonal volume, freshwater, and heat transport along the Indian coast**

**We thank the reviewer for the comments that helped us improve this manuscript. Given below are detailed replies to each of their comments. The reviewer's comments are in black and our responses are in red. The text from the manuscript, added or modified, can be identified by blue in italics and quotation marks.**

This is a very well-documented and very well-written paper. It concerns the application of a high-resolution model in the Northern Indian Ocean (NIO) with the view of revealing important patterns of the volume, salt, and heat transport under the prevailing climatological conditions. Apart from the direct outcome of this application, it also provides a quite interesting comparison with the findings of other numerical models in the region and available real-life datasets which is important and an asset of the paper. As a result, a pragmatic picture and features of the existing circulation and local physical phenomena are produced, most of which agree at a considerable degree with those described and analyzed by other researchers as well.

A general comment though is that the paper omits a discussion section but incorporates it instead in the results. Given that the authors devote little space on the affiliation and contribution of their results to the environmental problems of the NIO region (e.g., ecological, and biological processes, nutrient and carbon transport, fisheries etc.), I would recommend adding a separate section and elaborate more on the matter, so that the scientific contribution of this paper's findings to real life problems can be better stressed.

The authors thank the reviewer for the encouragement. As suggested by the reviewer, the results and discussion will be reported as separate sections. Regarding the biophysical relationships in the north Indian Ocean, we are already in the process of finishing an interesting article and communicating it in a suitable journal very soon.

Other than that, only some minor comments follow:

Line 59 'augmenting its physical model to extend to various biogeochemical processes' This doesn't read very well because this is a numerical model that implements the laws of physics but does not involve any physical modelling. Please rephrase.
Answer: This suggestion has been addressed. We have rephrased the line to: *'This model includes several built-in packages, facilitating the ability to simulate various biogeochemical processes, thereby advancing our understanding of complex oceanic interplays'*

Line 72 INC-HYC is only explained in section 2.1.2 and line 124 but the acronym needs to be determined here.
Answer: This has been addressed and line 72 has been rewritten as:

*'.... we conducted a model-to-model comparison of transports using another assimilated model data'.*

INC-HYC has also been explained at its first appearance in the manuscript.

Line 85 Please rephrase to 'this study aims further to help'
Answer: The suggestion has been incorporated.

Line 88 This sentence sounds like a phrasal error. Please rewrite it.
Answer: The line has been rewritten as *'In Section 3, we present the results and discussion'*.

Line 94 Refer to "Fig. 1 " inside the parentheses
Answer: This suggestion has been addressed.

Line 123 what do you mean by simulated? The model is run, built or setup at INCOIS?
Answer: The HYCOM model configuration is set up and maintained at INCOIS, which includes the T-SIS data assimilation scheme.

Line 131 ETOPO-1 acronym is not explained.
Answer: The line has been rewritten as *'A combination of GEBCO and Earth TOPOgraphy One Arc-Minute Global Relief Model (ETOPO-1) is used as the model bathymetry'*.

Line 139 NRL-HYCOM, INCOIS-GODAS. Please explain acronyms NRL and GODAS.
Answer: We have addressed this suggestion by adding the acronyms Naval Research Laboratory (NRL)-HYCOM and INCOIS-Global Ocean Data Assimilation System (GODAS) in the revised manuscript.

Line 515 similar comment to that for line 59 regarding the use of the term 'physical model'
Answer: The word 'physical' has been removed from the line in the revised manuscript.

Figure 1. The purple color does not work well in contrast to the red symbols. Change the color for the ADCP locations so that they can be better distinguished. There is a purple symbol missing from the Sri Lanka transect. Remove the brackets (in color scale). Please indicate where is AS and where BoB on the map.
Answer: The suggestions have been addressed and figure 1 has been revised.

Figure 5 Add units in the stick plots.
Answer: To make the validation section of the manuscript more concise, which was also suggested by the first reviewer, we have removed Figure 5 and its description from the revised manuscript.